# Wearable neurofeedback acceptance model for students' stress and anxiety management in academic settings

Sofia B. Dias[ORCID][1]*, Herbert F. Jelinek[2], Leontios J. Hadjileontiadis[3,4]

1 Interdisciplinary Centre for the Study of Human Performance (CIPER), Faculdade de Motricidade Humana, Universidade de Lisboa, Lisbon, Portugal, 2 Department of Medical Sciences, Khalifa University of Science and Technology, Abu Dhabi, UAE, 3 Department of Biomedical Engineering and Biotechnology; Healthcare Engineering Innovation Group (HEIG), Khalifa University of Science and Technology, Abu Dhabi, UAE, 4 Department of Electrical & Computer Engineering, Aristotle University of Thessaloniki, Thessaloniki, Greece

* sbalula@fmh.ulisboa.pt

## Abstract

This study investigates the technology acceptance of a proposed multimodal wearable sensing framework, named mSense, within the context of non-invasive real-time neurofeedback for student stress and anxiety management. The COVID-19 pandemic has intensified mental health challenges, particularly for students. Non-invasive techniques, such as wearable biofeedback and neurofeedback devices, are suggested as potential solutions. To explore the acceptance and intention to use such innovative devices, this research applies the Technology Acceptance Model (TAM), based on the co-creation approach. An online survey was conducted with 106 participants, including higher education students, health researchers, medical professionals, and software developers. The TAM key constructs (usage attitude, perceived usefulness, perceived ease of use, and intention to use) were validated through statistical analysis, including Partial Least Square-Structural Equation Modeling. Additionally, qualitative analysis of open-ended survey responses was performed. Results confirm the acceptance of the mSense framework for neurofeedback-based stress and anxiety management. The study contributes valuable insights into factors influencing user intention to use multimodal wearable devices in educational settings. The findings have theoretical implications for technology acceptance and practical implications for extending the usage of innovative sensors in clinical and educational environments, thereby supporting both physical and mental health.

## Introduction

Globally, mental health covers a spectrum from optimal well-being to profound suffering, affecting an estimated 970 million people, or one in eight individuals, with anxiety and depression being the most prevalent disorders [1]. Within the domain of higher education, mental health challenges among students have occurred over the past decades, indicating a significant public health concern [2, 3]. Research indicates a substantial increase in the prevalence of

**Data Availability Statement:** Data cannot be shared publicly because of legal restrictions imposed by the data sharing policy of Khalifa University. Data are available from an institutional contact point for researchers who meet the criteria

for access to confidential data and for research purposes only. For access, please contact the Ethics Committee at Khalifa University, via Dina Muhieddine (email: dina.muhieddine@ku.ac.ae).

**Funding:** HFJ was supported by Khalifa University of Science and Technology, under Grant FSU-847400341. The funders did not play any role in the study design, data collection and analysis, decision to publish, or preparation of the manuscript.

**Competing interests:** The authors have declared that no competing interests exist.

psychological distress upon entering university, with a rising demand for services addressing student mental health [4]. A systematic review and meta-analysis conducted by Steel et al. [5] also revealed that there is clear evidence of regional variation in the prevalence of common mental disorders; for example, while North and South East Asian countries exhibit lower prevalence estimates, regions such as the Middle East, affected by conflict, face heightened risks of post-traumatic stress disorder (PTSD), depression, and anxiety [6]. Studies in the Kingdom of Saudi Arabia and the United Arab Emirates (UAE) underline the prevalence of mental health issues among students, emphasizing the urgent need for research in this domain [7, 8].

At the same time, the COVID-19 pandemic has intensified existing mental health challenges, particularly for higher education students, adding an unpredictable layer of stress to an already concerning situation. The pandemic has amplified the need to address mental health among students, a concern identified well before the pandemic [9]. Numerous global studies during the pandemic shed light on the various impacts on the mental well-being of higher education students, revealing heightened levels of anxiety, stress, and depression (e.g., [10–12]). For instance, Chinese students reported a 45% prevalence of mental health problems, predominantly anxiety [13]. A 2020 multinational study involving nine countries found high prevalence rates of stress (61.3%), depression (40.3%), and anxiety (30%) among higher education students [14]. In the United States, stress and anxiety levels reached 71% [15]. UK surveys revealed high levels of anxiety and depression, increasing with sedentary behaviour and poor sleep quality [16, 17]. UAE students, specifically, demonstrated anxiety levels ranging from mild to severe in more than 50% of cases [18]. These studies highlight the critical importance of addressing mental health concerns in the context of higher education, requiring innovative approaches.

From the aforementioned, it is clear that efficient identification and management of student anxiety and stress are vital, having attracted the interest of many research groups proposing various sensing and intervention approaches, combined with technology acceptance models. This exploration aligns with the co-creation approach, emphasizing collaboration and multidisciplinary perspectives. The subsequent sections explore proposed approaches, integrating technology acceptance models, and highlight successful models that have produced positive outcomes in alleviating stress and anxiety among higher education students.

## Related work

The growing capabilities of wearable devices present opportunities for monitoring and addressing mental health challenges in the educational technology field. Wearables, equipped with diverse sensors such as inertial, physiological, and ambient sensors, demonstrate potential applications in mental health solutions (e.g., [19, 20]). Multimodal sensing, using multiple sensors simultaneously, has proven effective in various applications [21, 22]. Wearables can offer personalized and just-in-time services by considering contextual information about users. In fact, analyzing large amounts of sensor data is considered a complicated task, however, by using Artificial Intelligence (AI) and machine/deep learning (ML/DL) methods, it is possible to extract and interpret meaningful information from sensor data and use it to continuously monitor the current mental state of the user [23]. Researchers have explored the combination of bio-sensors to assess stress, providing examples such as multimodal emotion evaluation, driver anxiety detection, identification of cognitive tasks, and assessment of mental workload [24–28]. Moreover, a comprehensive overview about pain and stress detection using available wearable sensors was recently performed by Chen et al. [29]. In addition, different approaches have been explored for stress monitoring using mobile Electroencephalogram (EEG) head set MindWave [30], Electromyogram (ECG) and Electrocardiogram (EMG) [31], using a

combination of MindWave EEG, Zephyr BioHarness 3 chest belt, Shimmer Sensor [32], and mobile sensors suite AutoSense [33].

Stress management interventions are often classified based on theoretical backgrounds. Cognitive behavior therapy (CBT), third-wave (TW) concepts, mindfulness-based (MB) applications, and skills training (ST) are recognized categories [34–36]. A recent systematic review suggests that CBT, TW, and MB interventions yield better outcomes than ST programs [36]. Of particular interest is EEG biofeedback (neurofeedback) as an intervention for anxiety and stress management [37]. Neurofeedback enables users to self-regulate brain function, and studies show its effectiveness in reducing stress levels, especially in populations such as those with PTSD [38]. The application of EEG neurofeedback, such as alpha training, aims to induce relaxation [39]. Despite the plethora of existing mobile apps in this field, their suitability for addressing mental health in educational contexts remains uncertain.

The Technology Acceptance Model (TAM) is a widely used framework for evaluating user acceptance of general technologies, including those in educational technology [40–42]. Originally proposed by Davis [43], TAM considers that perceived usefulness and perceived ease of use influence users' attitudes and intentions towards technology use. TAM is based on social-cognitive models, such as the Theory of Planned Behavior [44], Diffusion of Innovations Theory [45], and Social Cognitive Theory [46]. Studies have demonstrated TAM's predictive value in health and technology contexts [47]. Modified TAM models, such as Unified Theory of Acceptance and Use of Technology (UTAUT) model, include additional predictors, for instance social influence and self-efficacy [48]. More recent modifications, such as mTAM [49], incorporate perceived novelty and perceived personalization constructs to explain user perspectives when adopting AI-based applications for behavior change. However, few studies have specifically examined TAM predictors related to mental health conditions (e.g., [50]). In fact, studies that tested specifically TAM predictors regarding acceptance and intention to use innovative devices for student stress and anxiety management are lacking. The most recent studies mainly focus on the usability of wearable headsets [51] or use just qualitative approaches to examine the acceptance and usability of particular technology (e.g., [52]), before studying the students'/stakeholders' perceptions related to the design of a specific device/conceptual approach and their intention to use.

Motivated by the identified gaps, the present study introduces a novel multimodal wearable sensing framework, named mSense, tailored for student stress and anxiety management in educational settings. The mSense framework includes modules for data acquisition, transmission, processing, and analysis. In addition, the study makes two main contributions: firstly, the design and development of mSense, leveraging biosignals to explore valuable information about student stress and anxiety levels not fully explored in existing educational technology research; secondly, the application of the TAM combined with qualitative content analysis to understand stakeholders' beliefs and perspectives related to student stress management. The findings offer insights for the successful design and adoption of mSense, providing implications for the educational technology field and strategies to transition non-adopters into adopters. The paper outlines the conceptual architecture, design implementation, TAM framework, hypotheses development, research method, data analysis, results, and a discussion on theoretical and practical implications, concluding with limitations and suggestions for future research in the context of educational technology and student well-being.

## Framework architecture and hypotheses development

### Architecture

The conceptual architecture of the mSense framework and corresponding modules are presented in Fig 1. As it is shown, the main sensing part of the mSense framework is a wearable

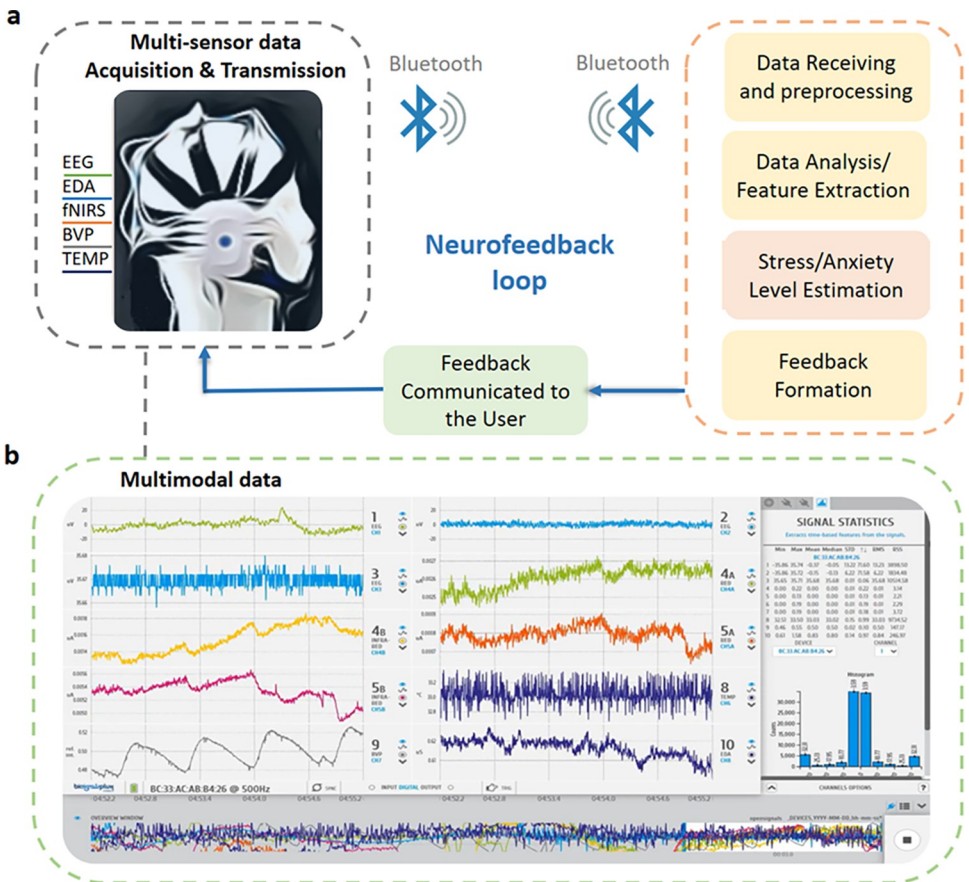

**Fig 1. Conceptual architecture of the mSense framework and corresponding modules.** (a) multimodal data acquisition/transmission, and (b) processing and analysis. EEG: Electroencephalogram (three channels); EDA: Electrodermal Activity (one channel); fNIRS: functional Near-Infrared Spectroscopy (two channels); BVP: Blood Volume Pulse (one channel); TEMP: Temperature (one channel).

headset that includes different sensors. In particular, real-time biosignals are sensed with a sampling frequency of 500Hz via three Electroencephalogram (EEG) channels, one Electrodermal Activity (EDA) channel, two functional Near-Infrared Spectroscopy (fNIRS) channels, one Blood Volume Pulse (BVP) channel, and one Temperature (TEMP) channel (see Fig 1 (A)). The EEG electrodes are placed according to the 10/20 international system, employing monopolar at Cz site and bipolar recording, where, in the latter case, the potential difference is measured between electrodes at the sites of Fp1/Fp2 snd T3/T4 (or C3/C4). The reference electrode is placed behind the right earlobe. The sampling frequency allows for EEG spectrum up to 250Hz which covers all the related EEG bands. The fNIRS electrodes are place bilaterally at the forehead (3 cm from the Fp1 and Fp2 sites, respectively). The BVP sensor is placed with a clip on the ear, and the TEMP sensor is mounted on the forehead between the Fp1 and Fp2 locations. The EDA sensor is placed on the index (+) and middle finger (-) of the right hand. The acquired data are transmitted via Bluetooth to a data receiver module that preprocesses the data, in terms of denoising and standardization. Then, a data analysis module extracts the features that relate with the body reaction to different stress and anxiety levels and feeds the module of stress and anxiety level estimation. The latter informs the feedback module that presents the corresponding feedback to the user, forming a neurofeedback loop (Fig 1(A)). In

Fig 1(B), a visualization of exemplified captured signals, i.e., three EEGs, two fNIRS, one EDA, one BVP, and one TEMP, is presented.

The selected types of biosignals are well connected with the stress and anxiety levels. In particular:

- EEG, a common signal for brain activity, is widely used in stress detection and evaluation [53]. Changes in EEG frequency bands (Delta (0.5–4 Hz), Theta (4–8 Hz), Alpha (8–13 Hz), Beta (14–30 Hz), and Gamma (30-80Hz)) correlate with stress levels. Altered rhythms, like increased delta in challenges, elevated theta in stress, suppressed alpha, and varied beta in task difficulty, offer insights into mental stress and psychological disorders [54].

- fNIRS indirectly measures brain activity by tracking haemoglobin concentrations. Changes in the prefrontal cortex, revealed by fNIRS, reflect stress-induced alterations in neural activity, making it useful for assessing stress effects [55].

- EDA sensors monitor skin conductivity changes, indicating stress-induced sweat reactions linked to the sympathetic nervous system. Responsive across contexts, EDA registers immediate increases in heart rate, respiration, and EDA during acute stress, with the sympathetic system, and parasympathetic system countering during post-stress recovery [56].

- BVP reflects the amount of blood travelling past a point along the vascular system at certain time intervals and is an indicator of blood flow, which increases under stress and decreases during a calm state [57].

- Under acute stress, sympathetically-mediated vasoconstriction causes a rapid drop in skin temperature; hence TEMP signals dynamic changes that can be used to reflect acute stress effects [58].

Next, examples of design implementation choices for the mSense module are described.

**Biosignals sensing module.** The biosignalsplux Hybrid-8 HUB from PLUX is employed as a biosignals sensing module in this study (https://www.pluxbiosignals.com). This 8-channel device digitizes signals from sensors, transmitting them via Bluetooth (Bluetooth & Bluetooth Low Energy support) to the data receiver (e.g., PC, smartphone), where they are recorded and visualized in real-time. OpenSignals PLUX software facilitates real-time signal visualization, as shown in Fig 1(B).

**Biosignals pre-processing and analysis modules.** Various preprocessing techniques enhance biosignal analysis, including down-sampling, bandpass filtering, and moving averages to remove baseline modulation and artifacts before normalization [59]. In particular, down-sampling reduces the sampling rate of the signal, which can help in reducing the amount of data and computational load, being particularly useful when the original sampling rate is higher than necessary for the analysis. The bandpass filters isolate the relevant components of the biosignal and remove the noise. For example, in ECG analysis, bandpass filters can remove low-frequency baseline wander and high-frequency muscle noise [59]. Moreover, moving averages technique smooths the signal by averaging a set number of data points. It helps in reducing random noise and making the underlying trends in the signal more apparent by removing baseline modulation and artifacts. Artifact removal techniques like Independent Component Analysis (ICA) [60] and adaptive filtering [61] are used to identify and remove artifacts from biosignals. ICA, for instance, can separate mixed signals into independent components, allowing for the removal of artifacts like eye blinks in EEG signals [62]. Finally, after removing noise and artifacts, normalization is performed to scale the signal to a standard range, making it easier to compare and analyze.

Feature extraction from preprocessed biosignals involves time-domain features such as mean normalization, root mean square, and means of differences between adjacent elements [63]. Advanced EEG/fNIRS features require diverse methods for association with mental stress [64]. More advanced signal decomposition approaches [65–69] enable the extraction of stress/ anxiety-relevant information, forming feature vectors for expert systems in estimation.

**Stress and anxiety level estimation module.** Various machine learning [70–72] and deep learning [73, 74] methods aim to recognize stress and anxiety levels. These approaches employ diverse techniques, including support vector machines, random forest, logistic regression, Bayesian networks, convolutional neural networks (CNNs), encoders, transformers, and recurrent neural networks (RNNs). To enhance representation, a multimodal fusion approach integrates features, models, and decisions from different biosignal modalities, enabling more accurate stress level recognition at feature-, model-, and decision-levels [75]. This strategy considers the multi-dimensionality of stress responses, facilitating effective classification and decision-making in stress and anxiety level assessment.

**Feedback formation module.** In neuro/biofeedback training, feedback enhances self-regulation, transferring skills into daily life [76]. Gaume et al. [77] highlight the importance of stress perceptibility, autonomy, mastery, motivation, and learnability in feedback for psychoengineering MB-based interventions. Perceptibility requires clear biosignal representation to reduce cognitive load. Autonomy shifts from external to internal feedback for self-regulation. Mastery and motivation focus on adaptable control, progress, and intrinsic motivation. Learnability emphasizes varied, engaging feedback for effective skill consolidation. Technological options, including visual, auditory, haptic methods, and virtual reality (VR), address these aspects, enhancing traditional MB-based interventions [78, 79].

## Hypotheses development

Taking into consideration the mSense TAM model, the present study adopts the "Perceived Usefulness" (PU), "Perceived Ease of use" (PEoU), and "Usage Attitude" (UA) constructs from TAM model, toward the model output of "Intention to Use" (IU), as depicted in Fig 2. To explore the contribution of each mSense TAM construct to the IU, a series of hypotheses (H1-H5) is introduced, as follows.

PEoU reflects the user's perceived easiness of the efforts needed to use the mSense framework, while PU reflects the perceived benefits that a participant can gain from using it. PEoU and PU have been widely adopted to predict technology adoption behaviors, showing significant effects on behavior intention [80].

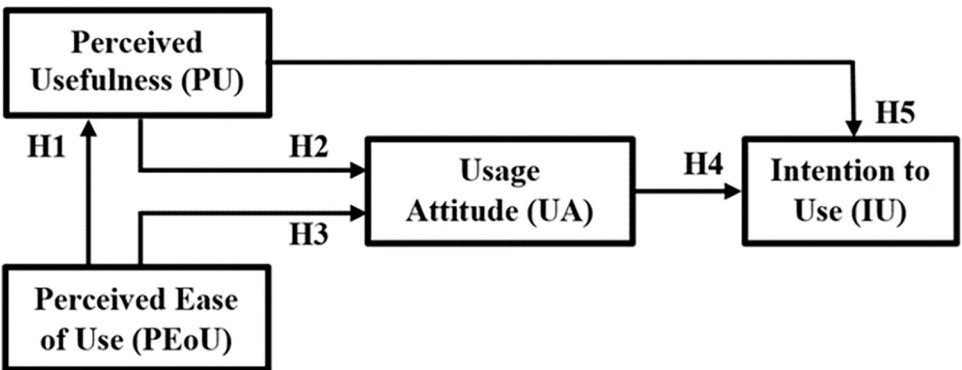

**Fig 2. The proposed mSense TAM framework and corresponding hypothesis (H1-H5).**

People usually tend to use or not to use a system or a technology based on the extent they believe it will help them perform their job better. Nevertheless, even if potential users believe that a given system is useful, they may, at the same time, feel that the system is too hard to use and that the performance benefits of usage are outweighed by the effort of learning and using the system. All else being equal, the easier it is to interact with a system, the less effort is needed to operate it, and so one can allocate more effort to other activities, contributing to overall job performance [43]. This provides the theoretical link between PEoU and PU, and the following hypothesis (H1) can be constructed:

**H1**: PEoU has a positive and significant relationship with PU.

Davis et al. [81] state that both PU and PEoU directly impact UA. Another study [82] mentions that the PU and the PEoU positively impact UA, revealing that when users are confident that the adoption of innovative technologies helps them to improve their work performance, they show more positive UA for adopting these innovative technologies. Moreover, if users perceive that do not need to apply much effort on the technology, then they have a more positive UA towards adopting this technology. From these perspectives, the following hypotheses (H2 and H3) are proposed:

**H2**: PU has a positive and significant relationship with UA.

**H3**: PEoU has a positive and significant relationship with UA.

In this study IU refers to individual willingness to use the mSense. Davis et al. [81] mention that UA directly influences IU, underlying that users' real usage behaviours are determined by IU, and their IU are determined by their individual UA. In this vein, it becomes clear that if the users have a more positive UA towards a particular system, then they will have a higher IU toward its use. Thus, we hypothesize the following (H4):

**H4**: UA has a positive and significant relationship with IU.

More recently, research performed by Farivar et al. [83] shown that by enhancing the perception of usefulness of the technology and emphasising its importance, there is a larger scope of the technology getting accepted by the end-users. In this line, the following hypothesis (H5) is proposed:

**H5**: PU has a positive and significant relationship with IU.

## Research method and data analysis

To effectively interpret the acceptance of the mSense framework, a multilevel analysis may be required. From this perspective, a convergent and exploratory triangulation mixed method was adopted to collect and analyze quantitative and qualitative data. In this way, different types of data were used, since the present study intended to find complementary data on the same subject and then merge the two sets of findings to produce an overall analysis [84]. In addition, while qualitative results may not be generalizable, mixed methods allow to hypothesize about (inter)relationships that may remain uncovered.

### Empirical study design

Between August and November 2023, a survey was conducted, inviting 300 mental health experts, doctors/health professionals, researchers, biomedical students, and designers/developers to provide feedback on their intention to use the mSense device. Email invitations were sent with a survey link using the Google Forms platform (https://forms.gle/qCR6CfW8UhX22aBC8). Convenience sampling was employed, selecting respondents based

**Table 1. Demographic profile of the participants ($n$ = 106).**

| Item | | Frequency | Percentage (%) |
|---|---|---|---|
| Gender | Male | 43 | 40.6 |
| | Female | 63 | 59.4 |
| Age | 18–24 | 44 | 41.5 |
| | 25–34 | 33 | 31.1 |
| | 35–44 | 15 | 14.2 |
| | 45–54 | 9 | 8.5 |
| | 55–64 | 3 | 2.8 |
| | Above 65 | 2 | 1.9 |
| Education level | High School | 23 | 21.7 |
| | Bachelors | 24 | 22.6 |
| | Masters | 25 | 23.6 |
| | PhD | 34 | 32.1 |
| Occupation | Researcher/Professor | 31 | 29.2 |
| | Software/Game Developer | 13 | 12.3 |
| | Healthcare Professional/Doctor | 10 | 9.4 |
| | High Education Students | 52 | 49.1 |

on ease of access. The survey, designed through a co-creation process [85], involving various stakeholders, aimed to be time-efficient (completed within 10 minutes), have a simple and objective structure, include demographic information, and gather feedback on user preferences. The final survey consisted of 12 questions with a mix of rating scale, multiple-choice, dichotomous, and open-ended formats. A total of 106 participants provided consent and completed the survey. The study received approval from Local Ethics Committees and Institutional Review Boards (Abu Dhabi, UAE, KU IRB approval ref. FSU-2021-001). Prior to enrolment, participants were provided with an online consent form that needed to be accepted to proceed with enrolment. The study excluded minors, as indicated by the demographics profile (Table 1).

The mSense was evaluated by one hundred six ($n$ = 106) volunteers. The data presented in Table 1 provide the demographic characteristics of the participants regarding the gender, age, education level, and occupation. The majority of the participants were female (59.4%) and were relatively young (18- 24yrs) (41.5%). Only 13.2% were older than 44 years old. The majority of the participants were PhD holders (32.1%) and high percentage of participants were higher education students (49.1%).

## Quantitative analysis

For testing the adopted mSense TAM framework, the Partial Least Squares-Structural Equation Model (PLS-SEM) technique was used, which is suitable for validating predictive models that use reflective latent constructs, without demanding big samples sizes and a priori data distribution assumptions [86]. PLS-SEM supports assessment of the measurement model and the structural one. The former is assessed via reliability and validity analysis, whereas the latter via the path significance and model predictive power analysis.

Reliability of mSense TAM constructs (see Fig 2) can be measured by Cronbach's alpha (CA) estimates, that assesses the unidimensionality of a block [87], the Dillon-Goldstein's coefficient ($\rho_j$), that focuses on the variance of the number of the blocks, and consistent reliability coefficient $\rho_\alpha$, as defined in Dijkstra and Henseler [88], that provides an estimate of the amount of obtained score variance that is due to true variance rather than to error. Additional

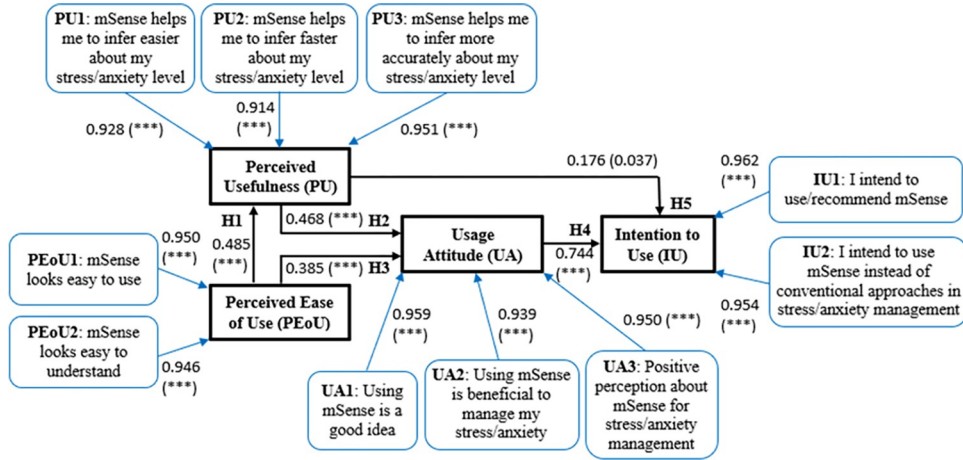

**Fig 3. Path loadings between the proposed mSense TAM main constructs and related items per construct.** H1-H5 denote the corresponding hypotheses (Fig 2); ***: $p < 0.001$.

metrics, such as Average Variance Extracted (AVE) and Composite Reliability (CR), evaluate overall variance captured by constructs and item sufficiency, respectively, with thresholds set at 0.70 (except AVE at 0.50) [89].

Convergent validity is assured if factor loadings are 0.7 or above and if each item loads significantly on its latent construct. Moreover, discriminant validity is assured when the following two conditions are met, namely: a) the value of the AVE is above the threshold value of 0.50, and b) the square root of the AVE is larger than all other cross correlations (namely Fornell–Larcker criterion). In addition, the discriminant validity of the constructs can be measured using the Heterotrait-Monotrait (HTMT) ratio of correlations for the adopted mSense acceptance model (see Figs 2 and 3), where all HTMT values should be below 0.90 [90].

The structural mSense TAM assessment is reflected in the path statistical significance and adj $R^2$ values, under a Bootstrap analysis. Additional structural assessment criteria were considered, such as the assessment of multicollinearity, the effect size $f^2$ and blindfolding-based cross-validated redundancy measure $Q^2$, and the out-of-sample predictive power of the model [89]. For the multicollinearity analysis, the Inner Variance Inflation Factor (VIF) is estimated and it should be less than 3.3, to indicate absence of lateral multicollinearity. The characterization of the effect size range is as follows: None: $f^2 < 0.02$, Weak (W): $0.02 \leq f^2 < 0.15$, Moderate (M): $0.15 \leq f^2 < 0.35$, Substantial (S): $f^2 \geq 0.35$. Furthermore, $Q^2$ establishes the predictive relevance of the endogenous constructs and values above zero indicate that the model has predictive relevance. Additionally, the model out-of-sample predictive power estimation is achieved via the comparison of the prediction root mean squared error (RMSE) between the PLS-SEM and the naive linear regression model (LM) benchmark [91]. For the interpretation of the model out-of-sample predictive power estimation process, the following characterization is adopted according to the number of item cases where the $RMSE_{LM} > RMSE_{PLS-SEM}$ holds, i.e., none: no predictive power, minority: low predictive power, majority: moderate to high predictive power, all: very high predictive power. Finally, a multi grouping analysis was performed to explore the TAM path coefficient effects from gender, age, education, and occupation.

## Qualitative analysis

The open-ended questions feedback from the online survey was used as follow-ups to the initial closed-ended questions, where the participant expands, in a freeform, his/her opinion(s).

Thematic analysis, utilizing a coding system, organized responses into meaningful categories, offering qualitative insights [92]. Two raters (first and third author) independently created (sub)categories related to mSense perspectives (e.g., usefulness, intention to use), ensuring a comprehensive analysis. Disagreements were accommodated via discussion between the raters, resulting in Cohen Kappa of 0.98. Examples of open-ended questions (OEQ1-OEQ4) include:

- OEQ1: Please specify why would you not use the mSense system to monitor and manage your stress/anxiety levels.

- OEQ2: Please specify any missing functionality/feature in the mSense system.

- OEQ3: Please specify any suggestion(s) that could improve the usability of this system.

- OEQ4: What is your opinion about the positive/negative consequences of such solution? Any other comments, improvements or issues related with this system would be most welcome!

## Statistical analysis

Stata version 17.0 and SmartPLS-SEM 4.0 (Partial Least Square Modelling) were used for the statistical analysis. Descriptive statistics were calculated for demographics and the related questions from the online survey. The statistical significance level was set to $p<0,05$, and the number of repetitions in the Bootstrap analysis was set to 200.

## Results

### Quantitative analysis findings

Fig 3 depicts the path relationships (loadings) between the proposed mSense TAM main constructs related with the adopted hypotheses (see Fig 2) and related items per construct. The identified items include: a) PU: {PU1: mSense helps me to infer easier about my stress/anxiety level; PU2: mSense helps me to infer faster about my stress/anxiety level; PU3: mSense helps me to infer more accurately about my stress/anxiety level}, b) PEoU: {PEoU1: mSense looks easy to use; PEoU2: mSense looks easy to understand}, c) UA: {UA1: Using mSense is a good idea; UA2: Using mSense is beneficial to manage my stress/anxiety; UA3: Positive perception about mSense for stress/anxiety management}, d) IU: {IU1: I intend to use/recommend mSense; IU2: I intend to use mSense instead of conventional approaches in stress/anxiety management}.

As it is clear from the path loadings of Fig 3, all present statistical significance ($p<0.05$). Moreover, Table 2 connects the path loadings depicted in Fig 3 with the measurement assessment outcome, in terms of the estimated Cronbach's $a$, DG $\rho_j$, $\rho_\alpha$, AVE, and CR values. As it is clear from Table 2, the threshold of 0.70 for all parameters but AVE, and the threshold of 0.50 for AVE are clearly met, with all estimated values overpassing 0.86. This indicates that the items used to represent the constructs (i.e., PU1-PU3, PEoU1, PEoU2, UA1-UA3, IU1, IU2) exhibit confirmed internal consistency reliability and convergent validity.

Additionally, Table 3 tabulates the results from the examination of the validity of the Fornell-Larcker criterion. As it can be seen from Table 3, this criterion is valid, as the square root of AVE in each construct (bold face) is larger than other correlation values among the constructs (column-wise). Furthermore, Table 4 shows the estimated HTMT ratio of correlations for the constructs of the adopted mSense TAM. As it is clear from Table 4, all HTMT values are less than 0.90, thus, ensuring discriminant validity between the reflective constructs. In addition, Table 5 presents the standardized path coefficients (with $p$ values) of the main

**Table 2. PLS-SEM, reliability, and validity analysis results for the adopted mSense TAM (see Figs 2 and 3).**

| PLS-SEM | | | Reliability Analysis | | | Validity Analysis | |
|---|---|---|---|---|---|---|---|
| Construct | Items | Loadings (*p* value) | Cronbach's *a* | DG $\rho_i$ | $\rho_a$ | AVE | CR |
| Perceived Usefulness (PU) | PU1 | 0.928 (***) | 0.923 | 0.951 | 0.923 | 0.867 | 0.951 |
| | PU2 | 0.914 (***) | | | | | |
| | PU3 | 0.951 (***) | | | | | |
| Perceived Ease of Use (PEoU) | PEoU1 | 0.950 (***) | 0.887 | 0.946 | 0.888 | 0.898 | 0.946 |
| | PEoU2 | 0.946 (***) | | | | | |
| Usage Attitude (UA) | UA1 | 0.959 (***) | 0.945 | 0.965 | 0.946 | 0.901 | 0.964 |
| | UA2 | 0.939 (***) | | | | | |
| | UA3 | 0.950 (***) | | | | | |
| Intention to Use (IU) | IU1 | 0.962 (***) | 0.911 | 0.957 | 0.916 | 0.918 | 0.957 |
| | IU2 | 0.954 (***) | | | | | |

*** $p < 0.001$

construct of Fig 3, extended with the corresponding path adjusted $R^2$ (adj $R^2$) values, under a Bootstrap fashion of 200 iterations. The latter range from 0.228 up to 0.752, indicating the significance of the adopted structure of the mSense TAM, justifying the validity of all adopted hypotheses, i.e., H1-H5 (Figs 2 and 3).

Furthermore, in Table 6 the estimated VIF is tabulated for the assessment of multicollinearity among the mSense TAM constructs. As all estimated VIF values are less than 3.3, no lateral multicollinearity exists among the mSense TAM constructs.

Moreover, Table 7 tabulates the estimated effect size $f^2$ between the mSense TAM constructs. From these results, it is clear that in most cases, the effect ranges from moderate (PEoU→UA, PEoU→PU) to substantial (PU→UA, UA→IU), except of the case between PU and IU, where the effect is weak.

Additionally, Table 8 shows the blindfolding-based cross-validated redundancy measure $Q^2$ for the endogenous constructs. The results in Table 8 indicate that all $Q^2$ values are higher than zero, justifying the predictive relevance of the proposed mSense TAM.

Table 9 tabulates the results from the model out-of-sample predictive power estimation. As is clear from Table 9, all estimated $Q^2$ values are greater than zero and the majority (5 out of 8) of dependent items indicators in the PLS-SEM analysis produced lower prediction RMSE ($RMSE_{PLS-SEM}$), when compared to the one from LM benchmark ($RMSE_{LM}$). This indicates that the model has moderate to high predictive power.

Finally, Table 10 summarizes the results from the multi grouping analysis that indicate (in boldface) any possible effect from the gender, age, education and profession on the main construct paths within the mSense TAM. Clearly, there is no effect from gender to all paths. A

**Table 3. Fornell–Larcker criterion: Inter-construct correlation vs. the square root of Average Variance Extracted (AVE) for the adopted mSense acceptance model (see Figs 2 and 3).**

| Constructs | Perceived Usefulness (PU) | Perceived Ease of Use (PEoU) | Usage Attitude (UA) | Intention to Use (IU) |
|---|---|---|---|---|
| Perceived Usefulness (PU) | **0.931** | | | |
| Perceived Ease of Use (PEoU) | 0.485 | **0.947** | | |
| Usage Attitude (UA) | 0.654 | 0.611 | **0.949** | |
| Intention to Use (IU) | 0.663 | 0.589 | 0.859 | **0.958** |

Off-diagonal elements are the construct correlations, while diagonal elements in bold and italic are the square root of the AVE.

**Table 4. Discriminant validity of the constructs based on Heterotrait-Monotrait (HTMT) ratio of correlations for the adopted mSense TAM (see Figs 2 and 3).** All HTMT values are below 0.90; hence, discriminant validity has been established between the reflective constructs.

| Constructs | Perceived Usefulness (PU) | Perceived Ease of Use (PEoU) | Usage Attitude (UA) |
|---|---|---|---|
| Perceived Ease of Use (PEoU) | 0.488 | | |
| Usage Attitude (UA) | 0.769 | 0.622 | |
| Intention to Use (IU) | 0.743 | 0.576 | 0.863 |

**Table 5. Standardized path coefficients ($p$ value) and adj $R^2$ values (Bootstrap analysis of 200 realizations).**

| Constructs | Perceived Usefulness (PU) | Usage Attitude (UA) | Intention to Use (IU) | Hypothesis (Accept/Reject) |
|---|---|---|---|---|
| Perceived Usefulness (PU) | - | 0.468 (***) | 0.176 (0.037) | H2 (A), H5 (A) |
| Perceived Ease of Use (PEoU) | 0.485 (***) | 0.385 (***) | - | H1 (A), H3 (A) |
| Usage Attitude (UA) | - | - | 0.744 (***) | H4 (A) |
| adj $R^2$ | 0.228 | 0.532 | 0.752 | |

***: $p < 0.001$

**Table 6. Multicollinearity assessment among the constructs via the Inner Variance Inflation Factor (VIF) estimation.** Note that all Inner VIF values are less than 3.3, indicating that lateral multicollinearity is not a concern in the adopted model.

| Constructs | Inner Variance Inflation Factor (VIF) | | | |
|---|---|---|---|---|
| | Intention to Use (IU) | Usage Attitude (UA) | Perceived Usefulness (PU) | Random Variable* |
| Usage Attitude (UA) | 2.041 | | | 2.817 |
| Perceived Usefulness (PU) | 2.041 | 1.238 | | 1.808 |
| Perceived Ease of Use (PEoU) | | 1.238 | 1.00 | 1.338 |
| Intention to Use (IU) | | | | 2.971 |

Last column shows the Inner VIF values after using the constructs as latent variables pointing a construct with a random variable, resulting in the Full Collinearity Test.

**Table 7. Effect size $f^2$.**

| Constructs | Usage Attitude (UA) | Perceived Usefulness (PU) | Intention to Use (IU) |
|---|---|---|---|
| Usage Attitude (UA) | - | - | 1.145 (S) |
| Perceived Usefulness (PU) | 0.652 (S) | - | 0.044 (W) |
| Perceived Ease of Use (PEoU) | 0.210 (M) | 0.238 (M) | - |

Effect size: None: $f^2 < 0.02$, Weak (W): $0.02 \leq f^2 < 0.15$, Moderate (M): $0.15 \leq f^2 < 0.35$, Substantial (S): $f^2 \geq 0.35$.

**Table 8. Predictive relevance of the model via the estimation of the $Q^2$ values.**

| Endogenous Constructs | $Q^2$ |
|---|---|
| Usage Attitude (UA) | 0.312 |
| Perceived Usefulness (PU) | 0.161 |
| Intention to Use (IU) | 0.255 |

**Table 9. Predictive model out-of-sample predictive power estimation via the comparison of the prediction root mean squared error (RMSE) between the PLS-SEM and the naive linear regression model (LM) benchmark.**

| Items | $Q^2$ | RMSE | | sign(RMSE$_{LM}$-RMSE$_{PLS-SEM}$) |
|:---:|:---:|:---:|:---:|:---:|
| | | PLS-SEM | LM | |
| PU1 | 0.153 | 0.923 | 0.917 | - |
| PU2 | 0.143 | 0.956 | 0.944 | - |
| PU3 | 0.104 | 0.947 | 0.951 | + |
| UA1 | 0.284 | 0.871 | 0.877 | + |
| UA2 | 0.206 | 0.948 | 0.945 | - |
| UA3 | 0.309 | 0.864 | 0.871 | + |
| IU1 | 0.261 | 0.901 | 0.906 | + |
| IU2 | 0.196 | 0.996 | 1.004 | + |

specific effect on PEoU→UA and PU→UA is noticed for the age differences {AGE1: 18-24yrs, AGE3: 35-44yrs} and {AGE2: 25-34yrs, AGE3: 35-44yrs}. The difference between the education level EDU1: High School and the rest (EDU2: Bachelors; EDU3: Masters; EDU4: PhD) affects mainly the paths UA→IU and PU→IU; the path PEoU→PU is affected by the specific differences in {EDU1, EDU3}, {EDU2, EDU3}, and {EDU3, EDU4}. Regarding the occupation, no effect is seen, apart from the difference between OCC1: Researchers/Professors and OCC4: Biomedical Engineering Students that affected the path UA→IU. These results indicate that: a) the low education level (EDU1) could affect the perception of the path from UA to IU, b) the way that PEoU leads to the PU could be affected by the difference in education level across all different education levels (EDU1-EDU4), and c) the distance between the occupation of researchers/professors from the biomedical students influences the perceived path from UA to IU. Clearly, these could be considered as quite specific effects that do not affect the generality of the mSense TAM framework, as shown from the results in Tables 2–10.

## Qualitative analysis findings

The qualitative analysis of the participant responses to open-ended questions related to facets of mSense (see Research method and data analysis-Qualitative analysis section), revealed interesting views. In comparison with other methods, many participants mentioned that the mSense framework is useful to manage stress and anxiety, and it can also be seen as a promising, unique, safe and unobtrusive solution potentially for long-term use. One of the students stated that:

*The mSense sounds really interesting and from my understanding it helps to detect levels of stress and anxiety.* [Participant 18, Undergraduate Student, OEQ4]

Another student underlined that:

*User can manage his/her anxiety and stress levels by dynamically interacting with the system, gaining insight about his/her performance. This helps him/her improve his mental condition in comparison to traditional management methods.* [Participant 38, Master Student, OEQ4]

In addition, one of students expressed that:

**Table 10. Multi grouping analysis, showing the path coefficient effects from gender, age, education, occupation.** 2-tailed *p*-values are included in brackets (boldface indicates statistically significant grouping effect, i.e., $p < 0.05$).

| Paths | Gender Diff. (F-M) | Age Diff. (AGE1-AGE2) | Diff. (AGE1-AGE3) | Diff. (AGE2-AGE3) | Education Level Diff. (EDU1-EDU2) | Diff. (EDU1-EDU3) | Diff. (EDU1-EDU4) | Diff. (EDU2-EDU3) | Diff. (EDU2-EDU4) | Diff. (EDU3-EDU4) | Occupation Diff. (OCC4-OCC3) | Diff. (OCC4-OCC1) | Diff. (OCC4-OCC2) | Diff. (OCC3-OCC1) | Diff. (OCC3-OCC2) | Diff. (OCC1-OCC2) |
|---|---|---|---|---|---|---|---|---|---|---|---|---|---|---|---|---|
| UA→IU | -0.128 [0.284] | -0.142 [0.491] | -0.116 [0.630] | 0.026 [0.881] | -0.583 [**0.025**] | -0.575 [**0.015**] | -0.787 [**0.002**] | 0.008 [0.945] | -0.205 [0.264] | -0.213 [0.129] | -0.145 [0.525] | -0.354 [**0.025**] | -0.082 [0.585] | -0.209 [0.453] | 0.063 [0.792] | 0.272 [0.056] |
| PEoU→UA | -0.254 [0.140] | 0.211 [0.895] | -0.498 [**0.017**] | -0.709 [**0.013**] | -0.169 [0.552] | -0.056 [0.828] | -0.336 [0.218] | 0.113 [0.730] | -0.167 [0.554] | -0.280 [0.333] | -0.510 [0.275] | -0.289 [0.096] | -0.294 [0.334] | 0.221 [0.549] | 0.216 [0.688] | -0.004 [0.850] |
| PEoU→PU | 0.212 [0.374] | 0.296 [0.604] | -0.049 [0.858] | -0.345 [0.784] | 0.152 [0.472] | 1.093 [**0.003**] | 0.179 [0.285] | 0.942 [**0.029**] | 0.027 [0.825] | -0.915 [**0.017**] | 0.024 [0.807] | 0.134 [0.584] | 0.888 [0.162] | 0.110 [0.608] | 0.864 [0.210] | 0.754 [0.294] |
| PU→UA | 0.244 [0.125] | -0.019 [0.221] | 0.442 [**0.034**] | 0.461 [**0.019**] | 0.178 [0.502] | 0.345 [0.144] | 0.379 [0.130] | 0.167 [0.508] | 0.201 [0.463] | 0.034 [0.907] | 0.728 [0.162] | 0.327 [0.070] | 0.508 [0.083] | -0.400 [0.383] | -0.219 [0.649] | 0.181 [0.591] |
| PU→IU | 0.083 [0.551] | 0.115 [0.230] | 0.050 [0.498] | -0.066 [0.219] | 0.595 [**0.022**] | 0.544 [**0.024**] | 0.794 [**0.010**] | -0.051 [0.832] | 0.199 [0.329] | 0.251 [0.171] | 0.001 [0.944] | 0.333 [0.053] | 0.022 [0.978] | 0.332 [0.353] | 0.021 [0.933] | -0.311 [0.170] |

M: Male; F: Female, AGE1: 18-24yrs; AGE2: 25-34yrs; AGE3: 35-44yrs (rest age ranges did not include significant number >9 to be included), EDU1: High School; EDU2: Bachelors; EDU3: Masters; EDU4: PhD, OCC1: Researchers/Professors; OCC2: Software Developers/Engineers; OCC3: Medical Doctors/Healthcare Professionals; OCC4: Biomedical Engineering Students.

*I think addressing treatment of stress and anxiety using such a system that targets these mental problems in terms of regulating the physiological signals generated by the sympathetic nervous system is a very unique and promising solution that could help in the long-term, non-invasive treatment of such mental problems with the right number of repetitions and regulations. I would personally rather prefer this approach of treatment than any other types of drugs that could negatively impact the biochemical pathways of my brain due to their unpredictable side-effects, while the use of this system is both non-invasive and safe to apply.* [Participant 65, Master Student, OEQ4]

Moreover, one of the medical doctors mentioned that:

*It is an innovative approach with direct implications for the participants.* [Participant 75, Medical Doctor, OEQ4]

Some participants also reported the potentiality and applicability of the mSense framework to other diseases/conditions (e.g., neurodegenerative diseases, stroke, psychological diseases). More specifically, one of the students stated that:

*Would be good to test the mSense framework even on the individuals with neurodegenerative disorders.* [Participant 50, PhD Student, OEQ4]

Another student underlined the following:

*I think it has great potential for people with anxiety disorders or people who are at high risk of a stroke and things alike. It's great for tracking what is normal and what is not and would definitely make the jobs of people in the medical field a lot easier, especially if they get access to the data from each user prior to their visit.* [Participant 13, Undergraduate Student, OEQ4]

Moreover, regarding the acceptability and related innovations aspects of the proposed mSense framework, many participants reported positive attitudes and intention to use/test it in future settings. For instance, one of the software engineers mentioned that the mSense framework is:

*Very interesting and innovative!* [Participant 67, Software engineer, OEQ4]

Additionally, two students stated the following:

*It is a non-invasive and user-friendly device.* [Participant 74, Undergraduate Student, OEQ4]

*It is a great system with multi functioning.* [Participant 48, Master Student, OEQ4]

The participants also provided some suggestions in order to improve some particularities of the proposed mSense framework. For instance, some participants emphasized the idea to have a mobile application for additional data visualization, to include less electrodes in the headset, to construct a user instruction manual regarding the use of the device, and others even suggested to design it more like a cap, in order to eventually extend its use outside home/clinical settings. In this line, three researchers suggested the following improvements:

*Connection with a mobile app for visualization of the data.* [Participant 33, Researcher, OEQ3]

*Make it like a cap.* [Participant 36, Researcher, OEQ3]

*Something to make the whole system to look more compact. Like a hat or a beanie (cap) connected to the electrodes so someone can wear it and outside of the house maybe. I am thinking something with less electrodes, by keeping only the necessary for stress and anxiety brain observation.* [Participant 46, Researcher, OEQ3]

In addition, two students added the following:

*Developing an app linked with the system that shows the level of stress and anxiety.* [Participant 19, Undergraduate Student, OEQ3]

*Making sure that a reader-friendly manual is provided to the user for reference at any time, as well as the availability of an experiment coordinator who could be reached out to at any time to help navigate the user through any necessary steps needed to use the system correctly, as the user might forget how to use the system at any given time.* [Participant 65, Master Student, OEQ3]

Additionally, some participants also suggested to integrate an extra functionality (brainwave music stimuli) in the headset related with the provided feedback and some participants underlined the importance to integrate in the future machine learning techniques in the conceptual design of the proposed framework, in order to better understand and interpret the related data and corresponding translation to general population, as follows:

*Instead of just giving stress level and anxiety level as feedback, it would be nice to provide another feedback to calm users down if anxiety/stress levels are high (e.g., calm music as in the muse headset).* [Participant 5, Software Engineer, OEQ2]

*I believe that the research in this area (neuronal activity) is very challenging, and it still has a great exploration margin. Machine Learning approaches can be a powerful tool to deeply understand some neuronal patterns linked with stress/anxiety, which would make this system a good ally for researchers and, ultimately, for the public.* [Participant 41, Software/hardware developer, OEQ4]

Moreover, some participants expressed concerns about adopting the mSense system, mentioning potential anxiety from brain signal tracking and how it might be perceived in public. Two students expressed specific limitations regarding their reluctance to use the mSense system:

"*I might use it once or twice, but I don't think I would wear the helmet regularly to collect this data.*" [Participant 28, Undergraduate Student, OEQ1]

"*Knowing that the brain signals are being tracked when I am anxious would eventually fuel my anxiety.*" [Participant 45, Undergraduate Student, OEQ1]

In addition to these concerns, other participants highlighted further considerations:

"*I believe I have a good understanding of my own mental state; yet, the headgear might provide additional insights.*" [Participant 78, Master's Student, OEQ1]

"*You might get funny looks on the train.*" [Participant 6, Researcher, OEQ1]

Overall, these perspectives highlight valuable feedback, touching on aspects such as practicality, social comfort, and emotional impact. While these insights point to areas for improvement, they also provide opportunities to refine the mSense system to enhance user-friendliness and accessibility for a broader audience.

## Discussion

mSense, a multimodal wearable sensing framework designed for managing stress and anxiety among academic students, is presented in this study. The exploration of its acceptance follows a co-creation approach, employing the Technology Acceptance Model (TAM). In the view of this approach, the described basic modules of mSense framework, i.e., multisensing, processing, analysis, stress/anxiety level estimation and feedback formation, provide an integrated approach in monitoring and applying MB-based interventions for student stress and anxiety management. The study places a particular emphasis on the intention to use mSense, examining both quantitative and qualitative results. The justified hypotheses (H1-H5) within the mSense TAM framework analyze the elements influencing users' intention to use mSense, shedding light on the constructs' interdependencies and their explanatory power in terms of usefulness, easiness, usage attitude, and usage intention.

Previous research shows that selecting an appropriate theory or model has always remained a critical task for researchers, since in the field of individual acceptance, different models and theories have been proposed (e.g., Theory of Reasoned Action (TRA), Social Cognitive Theory (SCT), Technology Acceptance Model (TAM), Theory of Planned Behaviour (TPB), Motivation Model (MM), Combined Technology Acceptance Model and Theory of Planned Behavior model (C-TAM-TPB), and Innovation Diffusion Theory (IDT)) [93]. Recently, Wang et al. [94] opted for the integration of C-TAM-TPB and IDT theories to investigate factors that affect university students' intention to adopt the Metaverse. In addition, a hybrid SEM-ANN analysis was performed by the authors, showing that subjective norm, attitude, compatibility, perceived usefulness, and relative advantages significantly affect Chinese university students Metaverse adoption intention, except for perceived behavioral control. Nevertheless, the selection of the suitable technology acceptance framework can be based on the availability of the variable(s) in the framework with evidence from previous studies that can be best predictors for the behavior intention, the validation of the framework (based on the popularity/citations), and the fecundity ("fertility" of the theory to generate new model and hypotheses) [95]. As Mardiana et al. [95] report, TAM exhibits the highest popularity and the most fecundity amongst the other technology acceptance frameworks. With regard to the evidence from previous studies, TAM was preferred in evaluating user adoption of technology-based solutions for stress/anxiety management [96, 97]. Based on this evidence, the TAM model has been adopted here to evaluate the technology acceptance of mSense framework.

Our results show that the mSense TAM, with its individual items and the related constructs (Fig 3) exhibits high reliability, composite, discriminant, and structural validity, along with predictive power. All hypotheses (H1-H5) were validated via the statistically significant path coefficients, confirming the robust construction of the TAM [43, 98]. More specifically, the path coefficient connecting PEoU with PU shows a positive and significant relationship (H1), which is in line with the findings reported by Estriegana et al. [99], Kang et al. [100], and An et al. [101]. The relationship between PU and UA (H2) was found to be also positive and significant, supporting the results of Estriegana et al. [99]. The path coefficient connecting PEoU with UA also reveal a positive and significant relationship (H3), in line with the findings of Estriegana et al. [99]. Furthermore, the relationship between UA and IU (H4) was found positive and significant, implying that participants accept and are indenting to use the mSense.

This relationship has been studied and found to be significant in various technologies, such as virtual worlds [102], online learning environments [99], academic social networking sites [103], online banking [104], and smart homes [105]. The results of the PU and IU also reveal a positive and significant relationship (H5), which is in line with the findings reported by Estriegana et al. [99], Kang et al. [100], and An et al. [101].

Apart from the quantitative results, qualitative data regarding the users' perspectives related with the mSense framework provided useful insights mostly related with the perceived usefulness and intention to use, as follows:

- most of the participants agreed that the mSense framework is useful for anxiety and stress management as a long-term solution; from the clinical point of view, it was mentioned that the mSense framework can be potentially used to support other diseases, such as neurodegenerative diseases (PU); and

- from the innovation point of view, many participants mentioned their interest and intention to use such system in the future, due its multiple and novel functionalities and user-friendliness characteristics (IU).

In addition, some improvements were suggested by the participants. For instance, in order to use the mSense framework in the clinical environments, the participants referred the need to create a detailed manual that shows all the characteristics and functionalities of the mSense framework.

Qualitative findings extend beyond PU and IU constructs, offering a detailed understanding of technology acceptance. The study's triangulation and mixed-methods approach, combining quantitative and qualitative data, enhances the comprehensive analysis. This strategy, previously employed in stress and anxiety management studies, provides valuable insights into mSense's acceptance context, contributing to a holistic view of user perspectives [106].

The group analysis results (Table 10), did not reveal a systematic effect, rather than some specific one, from the gender, age, education and profession on the main construct paths within the mSense TAM. However, some works have linked different aspects of personality to different features of persuasive technologies [107] and social cognitions (e.g., barriers, attitudes, and behavioral control) being predictive of technology uptake outcomes [108], pointing to some potential dimensions for further incorporation within the mSense TAM framework. In fact, mSense connects the digital (data collection- and algorithm-based inference) with the physical space (proximity to the human body), providing users with feedback on their past and current performance, allowing them to identify and adjust potentially unhealthy (stressful) behaviors. This is in line with the Oinas-Kukkonen's [109] concept of behavior change support systems, defined as

*Socio-technical information systems with psychological and behavioral outcomes designed to form, alter or reinforce attitudes, behaviors or an act of complying without using coercion or deception.*

Here we need to consider the dimension of self-leadership [110], in which individuals control their own behavior, leading themselves to achieve goals by using certain behavioral and cognitive strategies. The study explores various behavior-focused, natural reward, and thought pattern strategies, such as self-observation and positive self-talk, elucidating their application in stress and anxiety management [111].

In conjunction with this, our study extends this behavioral dimension to incorporate an IT-based leadership strategy, exemplified by mSense users willingly adopting and internalizing stress management techniques facilitated by the wearable. This potentiates the interplay of

leadership strategies between the user and mSense, i.e., transferred leadership strategies, where users first follow the leadership strategies provided by the wearable. Nevertheless, over time, they may internalize some of the strategies, and thus no longer depend on the wearable, but rather devise strategies for more sustainable behavior change; hence, in some cases, technology can be an effective means to develop self-leadership capabilities [111]. From a social perspective, the use of the mSense wearable could evoke a social comparison that can reveal a behavioral self-leadership strategy [112], that creates a different type of observation of oneself, within the context of others, and therefore may motivate one to go above and beyond one's performance limits. This is boosted by the social character of the academic environment, considering the constant interaction of the students with the technology, as well. In this vein, an accountability buddy effect [112] is also evoked, that may help people persist in trying new habits for longer periods of time.

## Implications, limitations, and future directions

The mSense TAM framework offers a comprehensive model for understanding the pathways influencing users' intention to adopt multimodal wearable sensing technology, such as mSense. It serves as a foundational structure that can accommodate additional constructs, allowing a detailed understanding of wearable framework adoption from end-users' perspectives across different technologies, user types, and usage settings. This contribution is particularly valuable in the context of the emerging research topic of multimodal wearable sensing. Moreover, the mSense TAM framework provides actionable knowledge for designers, guiding their focus on specific characteristics that enhance acceptance and intention to use among end-users. The highlighted constructs show the importance of creating easy-to-use frameworks for efficient and quick decision-making in health behavior [113]. The integration of quantitative and qualitative approaches, utilizing Partial Least Squares Structural Equation Modeling (PLS-SEM) and user perspectives, enhances the theoretical contributions of the study.

In the context of neurotechnology, wearable sensing technology holds great potential for revolutionizing interventions in mental health [114]. However, challenges related to the return on investment (ROI) need strategic efforts from industry stakeholders [115]. Personalized marketing campaigns emphasizing the benefits of multimodal wearable sensing can address user uncertainties and improve the PU of these devices. Transparency in data usage and building trust with end-users are critical considerations. Additionally, the emergence of diverse platforms like the Metaverse and Innerworld provides new opportunities for neuro/biofeedback solutions, such as the mSense framework, to enhance user engagement in extended reality (XR) environments. Industries should track users' affordance to understand and predict performance, allowing for the effective integration of stress/anxiety management in both the real world and the XR context. Understanding affordance actualization can be a strategic advantage for industries developing future XR technologies [116].

From a practical perspective, our study offers insights for wearable sensing vendors, emphasizing factors that enhance user-friendliness and effectiveness. In the educational technology context, wearables play a significant role in influencing students' mental states, particularly within neurofeedback therapy for stress and anxiety management. The application of wearable brain-computer interfaces (BCI) in innovative scenarios, such as iPhone apps [117] and serious game-based approaches [118], showcases the potential of these technologies in educational settings. Wearable technologies have also been utilized to discern mindfulness states during meditation practices [119] and in systems like Sensorium [120] for integrating neuro- and biofeedback to enhance self-awareness and relaxation. Moreover, multimodal wearable devices

have the potential to offer an alternative route to clinical diagnostics by leveraging various physiological information in real-time in a non-invasive or minimally invasive manner. In education, this could contribute to early detection, diagnosis, and monitoring of student well-being, offering high-resolution historical records of individuals' health status [121]. The implications of the mSense TAM framework extend beyond technology acceptance to practical applications with significant relevance in educational technology and health interventions.

Despite the promising results and outcomes presented here, some limitations should be outlined. Firstly, the mSense framework considers IU as a dependent variable, not actual behavior, posing a potential mismatch. The assumption is based on the fact that IU is closely related to actual behavior [98] with the method closely shared by many studies conducted on technology adoptions. Nevertheless, the actual adoption behavior may not necessary be reflected by the IU; Wang et al. [122] reported weak links in their study. Future extensions should incorporate actual behavior constructs to closely observe mSense's impact on students' behavior. Second, the presented work did not focus on a specific usage scenario; based on different scenarios and circumstances, the perception for the same service could be different. While this may not necessarily affect the results, as the study's intention is to understand the intention of users to adopt mSense as a means for managing their stress and anxiety, the study provides some direction for future research.

Thirdly, potential biases from the geographical context (UAE) could limit generalizability, prompting the need for cross-national research. Additionally, the study's three attributes for understanding IU may exclude important factors, such as system characteristics, past adoption behavior, and psycho-social factors such as individuals' personality traits and how they influence the application of leadership strategies and the associated behavioral outcomes [111]. Finally, the current work did not portray how users' attitude can change over time [123]. Davis et al. [81] for example, revealed that ease of use of technology adoption was significant after fourteen weeks of implementation contradicting to their earlier findings. This seems quite natural as useful features cannot be expected from the technology at the initial stage. In this vein, future research can employ a longitudinal study across the use of mSense for a sufficient period of time, as it would allow the changes in users' stress and anxiety management to flourish. The existing literature predominantly addresses acute stress/anxiety in mental health management, often neglecting chronic stress/anxiety. Chronic stress/anxiety, more severe and potentially fatal than its acute counterpart, significantly impacts overall health [124]. Emphasizing the importance of early intervention, future studies should prioritize the early stages of mental illness, aiming for proper diagnosis and treatment before it becomes a permanent aspect of an individual's life. Moreover, the potential of machine and deep learning-based approaches for stress/anxiety classification [125, 126], especially when combined with wearable AI [127], provide future explorative research pathways.

## Conclusions

Consistent evidence highlights the enhanced performance and robustness achieved through the utilization of multimodal sensing modalities in decoding mental states and responses to real-life cognitive challenges. Furthermore, the conceptual design and development of a research-grade multimodal wearable system open avenues to explore diverse research problems in real-world environments. This approach aligns with the mSense paradigm introduced in this study, providing a framework for entering into uncharted territories within the cognitive research field. More specifically, the present exploratory mixed-methods study performed an in-depth exploration and examination of the factors that influence user's intention for using a novel and promising multimodal wearable device for students' anxiety and stress

management. This potentiates the usage of such multichannel technologies and innovative sensors in clinical practice and educational settings, towards students' technology-based mental health support.

## Acknowledgments

The authors would like to thank all participants who voluntarily participated in the acceptance evaluation study.

## Author Contributions

**Conceptualization:** Sofia B. Dias, Herbert F. Jelinek, Leontios J. Hadjileontiadis.

**Data curation:** Sofia B. Dias, Leontios J. Hadjileontiadis.

**Formal analysis:** Sofia B. Dias, Leontios J. Hadjileontiadis.

**Funding acquisition:** Herbert F. Jelinek.

**Investigation:** Sofia B. Dias, Leontios J. Hadjileontiadis.

**Methodology:** Sofia B. Dias, Leontios J. Hadjileontiadis.

**Project administration:** Herbert F. Jelinek.

**Resources:** Leontios J. Hadjileontiadis.

**Software:** Leontios J. Hadjileontiadis.

**Supervision:** Herbert F. Jelinek, Leontios J. Hadjileontiadis.

**Validation:** Sofia B. Dias, Leontios J. Hadjileontiadis.

**Visualization:** Sofia B. Dias, Leontios J. Hadjileontiadis.

**Writing – original draft:** Sofia B. Dias.

**Writing – review & editing:** Sofia B. Dias, Herbert F. Jelinek, Leontios J. Hadjileontiadis.

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
