## [Decision Letter · Decision Letter 0]

15 Jul 2024

PONE-D-24-19965Wearable neurofeedback acceptance model for students' stress and anxiety management in academic settingsPLOS ONE

Dear Dr. Dias,

Thank you for submitting your manuscript to PLOS ONE. After careful consideration, we feel that it has merit but does not fully meet PLOS ONE’s publication criteria as it currently stands. Therefore, we invite you to submit a revised version of the manuscript that addresses the points raised during the review process.

We look forward to receiving your revised manuscript.

Kind regards,

Manob Jyoti Saikia, Ph.D.

Academic Editor

PLOS ONE

Journal Requirements:

2. In the online submission form, you indicated that "Data are available from the corresponding author to researchers who meet the criteria for access to confidential data and for research purposes only."

Reviewers' comments:

Reviewer's Responses to Questions

**Comments to the Author**

1. Is the manuscript technically sound, and do the data support the conclusions?

Reviewer #1: Yes

2. Has the statistical analysis been performed appropriately and rigorously? 

Reviewer #1: Yes

3. Have the authors made all data underlying the findings in their manuscript fully available?

Reviewer #1: No

4. Is the manuscript presented in an intelligible fashion and written in standard English?

Reviewer #1: Yes

5. Review Comments to the Author

Reviewer #1: Manuscript is well written. However some informations are found to be missing. The work incorporates EEG & other biosignals to develop the neuro feedback system. The sampling rate of acquired signals, location of placing electrodes, frequency band of EEG used for the neuro feedback system are unknown. Preprocessing of biosignals is not given much importance in the manuscript. Temperature is also considered for the feedback system. But there is no information how body temperature relates stress or anxiety. Secondly, the work uses people from different age groups. Number of participant per group is not available. But output responses are average out. Age might be an important factor when stress/anxiety is quantised. Thirdly, how the level of stress correlates with the TEM model?

Overall representation is found to be good.

6. PLOS authors have the option to publish the peer review history of their article (what does this mean?). If published, this will include your full peer review and any attached files.

Reviewer #1: No

---

## [Author Response · Author response to Decision Letter 0]

26 Jul 2024

Please refer to the attached: 

PONE-D-24-19965_Response to Reviewers

PONE-D-24-19965R1_Revised Manuscript with Track Changes

PONE-D-24-19965R1_Revised Manuscript

---

## [Decision Letter · Decision Letter 1]

3 Sep 2024

PONE-D-24-19965R1Wearable neurofeedback acceptance model for students' stress and anxiety management in academic settingsPLOS ONE

Dear Dr.  Dias,

Thank you for submitting your manuscript to PLOS ONE. After careful consideration, we feel that it has merit but does not fully meet PLOS ONE’s publication criteria as it currently stands. Therefore, we invite you to submit a revised version of the manuscript that addresses the points raised during the review process. One of our reviewers found a few more issues with the manuscript. Please address these issues and submit the revised version. 

We look forward to receiving your revised manuscript.

Kind regards,

Manob Saikia, Ph.D.

Academic Editor

PLOS ONE

Reviewers' comments:

Reviewer's Responses to Questions

**Comments to the Author**

1. If the authors have adequately addressed your comments raised in a previous round of review and you feel that this manuscript is now acceptable for publication, you may indicate that here to bypass the “Comments to the Author” section, enter your conflict of interest statement in the “Confidential to Editor” section, and submit your "Accept" recommendation.

Reviewer #1: All comments have been addressed

Reviewer #2: (No Response)

2. Is the manuscript technically sound, and do the data support the conclusions?

Reviewer #1: Yes

Reviewer #2: No

3. Has the statistical analysis been performed appropriately and rigorously? 

Reviewer #1: Yes

Reviewer #2: Yes

4. Have the authors made all data underlying the findings in their manuscript fully available?

Reviewer #1: Yes

Reviewer #2: Yes

5. Is the manuscript presented in an intelligible fashion and written in standard English?

Reviewer #1: Yes

Reviewer #2: Yes

6. Review Comments to the Author

Reviewer #1: Review comments are well addressed. The manuscript covers all the details of the experiment and the results are well explained.

Reviewer #2: The manuscript investigates the acceptance of a conceptual multimodal wearable sensing framework, mSense, which aims to give real-time neurofeedback for stress and anxiety management in students. As such, they distributed an online survey to individuals above 18 years of age to assess the perceived usefulness (PU), perceived ease of use (PEoU), usage attitude (UA), and intention to use (IU) on this conceptual technology. While most of the manuscript is well written, I feel there is a major problem with the Study Design section. There is very little information about the participants, their background in multimodal biosensor or wearable tech, and how much - if any - briefing they received about the proposed conceptual technology prior to completing the survey. Highlighted below are the major and minor points I would like to see addressed:

1. P 5, L127: Please provide a citation for mTAM.

2. P6: I understand that this is a conceptual framework, but as a researcher who does multimodal biosensor work, I wonder how feasible it is to set up, connect, and synchronize all of these biosensors without professional help. This hugely affects usability and PEoU and potentially IU. Please include technical feasibility of the application/use of this device by novice users. Please also indicate whether the participants were briefed on the setup of the device as it is crucial in understanding whether they were able to effectively evaluate PEoU, UA, and IU.

3. P 6, L163: clip in the ear should be clip on the ear, I presume?

4. Given that their analysis is based on participants evaluating a conceptual framework, I would have liked more details about the types of feedback given to the individuals, how it is delivered, when is it delivered (do they only receive feedback when they feel stress/anxiety?), if there are any delays (as there usually are in most types of real-time neurofeedback). Did the participants receive any details about the feedback given? (I did not see any information on the survey).

5. Page 11, L 280: They state that they invited 300 mental health experts, doctors/health professionals, researchers, biomedical students, and designers/developers to participate in the survey research. Were all participants familiar with all of the biosensors? I looked at the survey and there is no description of any of the biosensors used. How do we know that they understand what these biosensors do and how to set them up successfully and use them? If they do not know how to use these biosensors, how could they rate their PEoU, UA, or IU?

6. Page 12: Please provide upper range of participants’ age.

7. Page 19~: It would be helpful if the authors broke down which of these qualitative responses were in response to which of the four open ended questions. For example, which ones were in response to the first open ended question: Please specify why would you not use the mSense system to monitor and manage your stress/anxiety levels.

7. PLOS authors have the option to publish the peer review history of their article (what does this mean?). If published, this will include your full peer review and any attached files.

Reviewer #1: No

Reviewer #2: No

---

## [Author Response · Author response to Decision Letter 1]

15 Sep 2024

Please refer to the response letter to the reviewer's comments.

---

## [Editor Report · Decision Letter 2]

7 Oct 2024

Wearable neurofeedback acceptance model for students' stress and anxiety management in academic settings

PONE-D-24-19965R2

Dear Dr. Dias,

We’re pleased to inform you that your manuscript has been judged scientifically suitable for publication and will be formally accepted for publication once it meets all outstanding technical requirements.

Kind regards,

Manob Saikia, Ph.D.

Academic Editor

PLOS ONE

---

## [Editor Report · Acceptance letter]

15 Oct 2024

PONE-D-24-19965R2 

PLOS ONE

Dear Dr. Dias, 

I'm pleased to inform you that your manuscript has been deemed suitable for publication in PLOS ONE. Congratulations! Your manuscript is now being handed over to our production team.

Kind regards, 

on behalf of

Dr. Manob Jyoti Saikia 

Academic Editor

PLOS ONE